# Substance Use and Housing Stability among Individuals Experiencing Homelessness and a Traumatic Brain Injury: The Role of Social Support

Stephanie Chassman [1],*, Grace Sasser [1], Sara Chaparro Rucobo [2], Katie Calhoun [1], Anamika Barman-Adhikari [1], Blair Bacon [3], Kim Gorgens [2] and Daniel Brisson [1]

[1] Graduate School of Social Work, University of Denver, 2148 S. High St., Denver, CO 80208, USA
[2] Graduate School of Professional Psychology, University of Denver, 2450 S. Vine St., Denver, CO 80210, USA; sjchaparror@gmail.com (S.C.R.)
[3] Anschutz School of Medicine, University of Colorado, 1890 N Revere Ct, Aurora, CO 80045, USA
* Correspondence: stephanie.chassman@du.edu

**Abstract:** Purpose: When compared to the general population, people experiencing homelessness have significantly higher rates of TBI (traumatic brain injury). Individuals experiencing homelessness and a TBI require social support because it can serve as a protective factor in reducing the risks of substance use and positively impact housing stability. This study aimed to better understand how social networks influence housing stability among individuals experiencing homelessness and a TBI. Materials and methods: A purposive sampling design was utilized to recruit and survey 115 adults experiencing homelessness. Quantitative questions captured data on demographic information, brain injury-related variables, homelessness-related variables, social network support types and characteristics, and correlates of housing instability including self-report substance use variables. Results: Findings showed that substance use was, indeed, a barrier to stay in or afford housing. Additionally, rates of social support were uniformly low across the sample, showing the unique vulnerabilities associated with homelessness and TBI and homelessness in general. Conclusion: Intervention efforts may consider fostering support networks, as social support has been linked to both housing stability and non-housing outcomes such as reduced substance use, improved health, and community reintegration.

**Keywords:** homelessness; traumatic brain injury; social support; substance use; housing stability

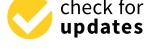



## 1. Introduction

### 1.1. Homelessness-Navigating a Precarious Existence

As of January 2020, an estimated 580,000 people were without shelter [1]. In 2020, homelessness had increased by 2% compared to 2019, marking the fourth consecutive year of increases in homelessness in the United States [1]. Causes of homelessness are complex and nuanced for each individual but often involve interactions between individual-level and structural risk factors [2–4].

Individual-level factors such as family conflict, mental health problems, substance use, early childhood adverse experiences, personal history of violence, and criminal justice system involvement [5–8] all contribute to increased risk of homelessness. Further, social and emotional problems, learning disabilities, memory lapses, and poor executive functioning as a result of neurological injuries (i.e., traumatic brain injury—TBI) might also be risk factors of homelessness [9,10]. Lack of affordable housing, loss of one's job, eviction, domestic violence, medical debt, lack of insurance, and income inequality [3,11,12] are thought to be structural level factors that have been linked to homelessness. Once homeless, individuals are often exposed to many other deleterious circumstances that

increase their vulnerability to detrimental health and mental health outcomes (i.e., toxic stress, victimization, and cognitive dysfunction including TBI) [13–15].

## 1.2. Traumatic Brain Injury (TBI) and Homelessness

TBI is defined as a neurological injury that impacts how the brain works [16]. There are various causes of TBI, including violence (e.g., assault), falls, motor vehicle accidents, substance-use related accidents, and sports. TBI can meaningfully impact a person's capabilities for self-regulation, planning and organizing, judgment, reasoning, and problem-solving; emotional and behavioral changes often occur [17]. Consequently, individuals with TBI history may suffer from impulsivity, mood swings, and personality changes [18], and this often impacts one's ability to maintain employment and stable housing.

Instances of TBI are higher among individuals who are unstably housed (over 50%) compared to the general population (12%) [9,14,19,20]. Literature has found over 60% of individuals experiencing homelessness with a history of TBI had experienced more than one TBI [9]. Other literature has found that in a quarter of the population of individuals experiencing homelessness with a TBI, the injury was moderate or severe; this finding is "10 times that of the general population" [21]. Individuals experiencing unstable housing are at a disproportionately high risk for sustaining a TBI due to some of the dangers of living on the street, including victimizations by assault and substance abuse-related injuries [22,23]. One study found homeless status to be highly correlated to head injuries [24]. Prior research may suggest that homelessness status may be a contributing factor in the increased rates of TBI, potentially due to the dangerous circumstances associated with street life.

In addition to the direct consequences of a TBI (i.e., lifelong cognitive impairments [25], individuals may also experience psychosocial difficulties such as unemployment, social isolation, relationship breakdown, and potentially homelessness, as individuals struggle to manage and come to terms with the functional impact of their injuries [26]. According to Hwang et al., (2008), [9], a history of TBI is strongly associated with many adverse health conditions among individuals experiencing homelessness, including seizures, poor mental health, and substance abuse problems.

### 1.2.1. Homelessness and Substance Use

Substance use is a common comorbid disorder among individuals experiencing homelessness [14,27]. Rates of substance use among individuals experiencing homelessness are consistently above average [17,28]. A meta-analysis found alcohol dependence ranged from 8.1% to 58.5% and drug dependence ranged from 4.5% to 54.2% among individuals experiencing homelessness [17]. Not only do individuals experiencing homelessness have higher rates of alcohol and drug use, but research also shows this vulnerable population to have greater severity of alcohol and drug use, leading to high rates of drug overdose compared to housed individuals [27,29]. A study among veterans experiencing homelessness found that substance use can decrease opportunities to establish and maintain housing and employment, and increase exposure to victimization [30]. While substance use may serve as a risk factor for homelessness, it may also serve as a barrier to transition from homelessness to stable housing [31].

### 1.2.2. TBI, Homelessness, and Substance Use

TBI among individuals in unstable housing has been linked to substance use. Research found support for a relationship between substance use and TBI [32]. Substance use increases the risk of housing instability and of TBI [9,33]. Additionally, substance use may have a negative impact on recovery from TBI [33].

Research has found that alcohol abuse [9,34,35], marijuana use, and crack or cocaine use were commonly used substances among individuals with a TBI [36]. Moreover, individuals with a reported TBI often also reported alcohol intoxication at the time of injury [33], further showing that substance use may be a risk factor of TBI.

### 1.3. Social Support-Attenuating the Risks Associated with Homelessness

Adequate social support can serve as a protective factor in reducing the risks of substance use among individuals experiencing homelessness. There is evidence that demonstrates that those who have a strong social support system have a higher chance of recovery from substance use and are less likely to return to substances in the future [37,38]. Not only is social support correlated with lower substance use, but social support has also been seen to benefit physical, mental, and emotional well-being [39–41]. Without adequate social support, individuals experiencing homelessness may remain disconnected from social services and have less support in navigating stressors, such as substance use and housing instability [38,42]. TBI may add additional barriers to maintaining social support, housing stability, and low substance use due to the consequences of a TBI, such as personality problems, behavioral issues, and social and intellectual problems [43].

### 1.4. Social Support among Individuals with a TBI

Individuals with a reported TBI may face strained interpersonal relationships, social isolation, and relationship breakdown [26,38,44], potentially leading to low levels of social support. Low social support was found to be associated with higher severity of alcohol abuse, higher likelihood of relapse following substance use treatment, and fatigue among TBI patients [37,45]. The consequences of a TBI, especially in conjunction with substance use, lead to impairments in physical, mental, emotional, and social functioning [26]. This, in turn, makes it difficult to develop and maintain positive social relationships. Without adequate social support, this highly vulnerable population may experience housing instability and homelessness, and may be less likely to access necessary medical care for TBI or substance use treatment [46,47]). Individuals experiencing homelessness and a TBI may face longer and more frequent periods of housing instability, lower levels of social support, and higher rates of substance use [26,37,48].

### 1.5. Social Network Composition among Individuals Experiencing Homelessness and a TBI

Research has identified the importance of developing positive social support as a means of exiting homelessness and increasing housing stability [40,41,46] because social support can help buffer the effect of stress on well-being [42]. Qualitative research has shown positive social support can help individuals experiencing homelessness find more stable housing, specifically when the support is provided by family or caseworkers [47,49]. Family support is also important for people with a TBI [43]. Sufficient social support from family, friends, and partners was shown to decrease the occurrence of long-lasting mental fatigue among individuals with a TBI [45]. However, for individuals experiencing homelessness, family relationships might be tenuous [50] as isolation and conflict with family members is often a contributing factor toward homelessness [51,52].

Street-based peers may serve as primary sources of social support among individuals experiencing homelessness because individuals experiencing homelessness often rely on each other for survival and resources [53]. However, street-based peers may not provide positive support; research has found that networks of street-based peers can increase distress among individuals experiencing homelessness and unhealthy behaviors [54]. More often, networks of street-based peers may be a negative influence on health-related outcomes, service use behavior, and greater dependence on the street economy [55–57]. Understanding social support may be particularly pertinent for people who are both experiencing homelessness and have a TBI, as social support has been found to be important for housing outcomes among this population [40,41,43,46].

While these aforementioned studies shed light on some of the social-network characteristics of individuals experiencing homelessness and individuals with a reported TBI, there are important limitations to consider. These studies did not utilize samples of individuals experiencing homelessness and a TBI. While these studies contributed to the literature in terms of examining the impact of social network characteristics, comparing social network characteristics among two groups (with and without a reported TBI) enables

researchers to understand with more specificity the impact that social networks can have on individuals [58,59].

This study will attempt to address this gap in the literature by examining the social network characteristics among individuals experiencing homelessness and a TBI in order to gain a better understanding of the influence of social networks on housing stability.

*1.6. Theoretical Framework*

1.6.1. Social Capital Theory

Social capital theory [60] provides a helpful and concise framework to understand how social networks influence housing stability among individuals experiencing homelessness and a TBI. Social capital has been defined as one's ability to accrue benefits by virtue of their personal relationships with others and by belonging to social networks [61]. Lin (1990, 1999), [62,63] explains social capital theory as an investment in social relations with expected returns, such as facilitating the flow of information, and views social capital as resources that are accessed through social ties. Lin et al., (2001) [64] focuses on the resources embedded within social networks and argues that not all resources are equal (i.e., social capital's impact on individual well-being is variable). An individual's social capital may differ depending on the type of network they have.

Research has used this theory among young adults and adults experiencing homelessness [55,65–70]. Results showed social capital was a significant correlate of service engagement, potentially leading to housing stability [65,71–74]. More specifically, mixed results highlight how social capital from street-based peers can have a positive or negative influence on service engagement, housing stability, and well-being [65,75]. Support from home-based peers and caseworkers, however, has been consistently found to have a positive impact on service engagement as a way to transition out of homelessness [76,77].

Although this theory has been applied to young adults and adults experiencing homelessness, their samples did not include individuals experiencing homelessness with a TBI. Specific to the TBI population, research has found that individuals who had stronger social support had lower levels of emotional distress and were more likely to be employed, in school, or in a training program [78,79]. These results demonstrate the importance of social support and positive social capital in assisting with recovery from TBI and maintaining employment. This study will further analyze and apply social capital theory to adults experiencing homelessness and a TBI to determine its applicability to this population.

1.6.2. Types of Support

There are three primary types of social support: instrumental support, emotional support, and service/informational support [80]. Instrumental support comes when social connections provide tangible help ([80] p. 5), such as delivering a meal or providing a place to stay for the night. Emotional support is defined as "instances in which the participant was able to confide in loved ones about problems and worries" ([80] p. 5). Emotional support is the most well-known type of support, as it is the type that people usually consider when we think of family and friend relationships [80]. Informational support, sometimes referred to as service support, encompasses the exchange of information ([81] p. 228) such as a case worker/social worker: someone who may provide a service in a time of need. Research has shown that among various social networks of individuals experiencing homelessness, unhoused and housed peers provide instrumental, emotional, and informational support to one another [53].

1.6.3. Bridging and Bonding Social Capital

The social ties that individuals keep span two kinds of social capital, bridging and bonding social capital [82]. Studies show that individuals experiencing homelessness who are able to access these two sources of social support have both positive and negative outcomes across domains including housing stability and retention [83–85].

Bridging social capital refers to the social support provided by heterogeneous networks, such as family members who are able to assist with housing [53] or service providers who are able to provide professional support after a TBI [86]. These relationships may expose individuals to information and resources [55] that may help distance themselves from the challenges of living on the streets and achieve housing stability.

Alternatively, bonding social capital refers to the social support provided by homogeneous networks (i.e., others who are experiencing homelessness) [53,67]. Studies show that these sources of social capital do not provide opportunities for mobility; rather, relationships with others in the same social position may be unstable [57,70]. These unstable relationships may decrease the likelihood of exiting homelessness [87], as support from other street-based peers may not provide many opportunities to transition out of homelessness [67].

1.6.4. The Current Study

This paper seeks to examine social networks among individuals experiencing homelessness and a TBI. This study will organize social network composition through the use of primary and secondary networks. The primary network typically fulfills most of the social support functions and consists of a romantic partner, family, and close friends [88,89]. On the other hand, the secondary network includes more formal relationships [90] such as with caseworkers, co-workers, and people from school. Research questions and hypotheses include:

(1) What differences may exist in social networks among individuals experiencing homelessness and a TBI compared to individuals experiencing homelessness only?

(a) We hypothesized that individuals experiencing both homelessness and TBI will have lower overall social support than individuals experiencing only homelessness due to the potential negative impact of TBI on social support.

(2) What is the relationship between types (instrumental, emotional, informational support) and sources (primary and secondary network) of social support among individuals experiencing homelessness and the impact on housing stability related to substance use?

(a) We hypothesized that social support from the primary network is associated with increased housing stability. Studies have found the beneficial effects of family members on housing stability [47,49,53,86]. Alternatively, social support from the secondary network will be associated with decreased housing stability, as evidenced by studies [57,66,67,70,87] demonstrating the potentially detrimental influence of street-based peers.

## 2. Materials and Methods

### 2.1. Study Setting

In 2020, researchers from The University of Denver (the Center for Housing and Homelessness Research and the Graduate School of Professional Psychology) partnered with two community organizations across Colorado serving individuals experiencing homelessness [91]. The dataset came from a two-site study (The Murphy Center for Hope in Fort Collins and Catholic Charities' Marian House in Colorado Springs) examining the relationship between TBI and homelessness [91]. Community partners supported the study by hanging recruitment flyers and encouraging service recipients to visit on the day of the data collection.

### 2.2. Sample and Recruitment

Purposive sampling was used to recruit and survey 115 English-speaking adults (ages 18–73). A standardized protocol for recruiting and screening potential participants was used. The eligibility screener asked if a participant was over 18 years old and experiencing homelessness. Written informed consent was given from eligible participants before beginning data collection.

Quantitative questions captured data on demographic information, brain injury-related variables, homelessness-related variables, social network characteristics, and correlates of housing instability including self-report substance use variables. The Institutional Review Board (IRB; 521142-9 and 19 October 2021) at the University of Denver approved all study procedures prior to data collection.

### 2.3. Data Collection Procedures

Participants were given the written consent form and asked if they would like to read it themselves or have it read to them. Once written consent was obtained, researchers read each survey question to participants and allowed participants to answer. Participants were informed they could skip any questions they were uncomfortable answering and trained staff was available for support. The survey took approximately 25 min to complete. Participants were given a $15 gift card to a local grocery store as compensation for survey completion [91].

Study data were collected and managed using REDCap electronic data capture tools hosted at The University of Denver. REDCap (Research Electronic Data Capture; ref. [92] is a secure, web-based application designed to support data capture for research studies, providing: an intuitive interface for validated data entry, audit trails for tracking data manipulation and export procedures, automated export procedures for seamless data downloads to common statistical packages, and procedures for importing data from external sources.

### 2.4. Measures

2.4.1. Sociodemographic Characteristics

Sociodemographic characteristics were controlled for in the analyses. They included the following variables. The study site was either Fort Collins or Colorado Springs. Gender was captured by three categories (male, female, other-specify). Sexual orientation was categorized into five categories (bisexual, gay, heterosexual, lesbian, not listed). Since a majority of participants identified as one of either male or female and heterosexual or LGB (99% of the sample identified as male or female and 85% of the sample identified as heterosexual), gender identity and sexual orientation were dichotomized (male or female; heterosexual and LGB and not listed), all other cases were dropped from analyses. Race/ethnicity was originally categorized into eight categories (American Indian/Alaska Native, Asian, Native Hawaiian or Other Pacific Islander, Black or African American, White, Hispanic, more than one race, unknown/not reported) and recoded into two categories (white and BIPOC) for analysis, due to a small sample in some racial categories (e.g., Asian, American Indian/Alaska Native). Educational attainment was measured using the following categories—less than a high school diploma, high school degree or equivalent, Associate degree, Bachelor's degree, Master's degree, Doctorate, Other—and then recoded into two categories (high school degree or less, and more than high school degree), similarly, due to a small sample size in some categories (e.g., Master's degree, Doctorate).

2.4.2. VI-SPDAT

The Vulnerability Index- Service Prioritization Decision Assistance Tool (VI-SPDAT; [93] is a standardized measure that was used to capture information on homelessness status. The VI-SPDAT is the homelessness status tool used by the Continuum of Care (COC) under the Department of Housing and Urban Development (HUD) to capture information on homelessness status to determine who should receive housing assistance first [93]. The VI-SPDAT was used to assess history of housing and homelessness, risk behavior, socialization and daily functioning, and wellness.

Substance Use Related Housing Stability (Dependent Variables)

The impact of substance use on housing stability was measured by the VI-SPDAT, examples of the questions included: "Has your drinking or drug use led you to being

kicked out of an apartment or program where you were staying in the past?" and "Will drinking or drug use make it difficult for you to stay housed or afford your housing?" All the responses were coded to 1 = yes and 0 = no.

### 2.4.3. OSU TBI-ID

The Ohio State University TBI Identification Method (OSU TBI-ID; ref. [94]) is a standardized screening measure which was used to capture data on brain injury. The OSU TBI-ID was used to collect information on participants' history of TBI [94] and information served as control variables. The OSU- TBI-ID is a standardized structured interview procedure designed to capture information on lifetime TBI histories. Participants are considered to have a significant history of TBI if they reported a "first" TBI with loss of consciousness (LOC) before age 15, a "worst" TBI with LOC longer than 30 min, or a "multiple" TBI event, defined as "a period where three or more blows to the head caused altered consciousness OR two or more TBIs with LOC within a 3-month period" ([95] p. 16). For analysis, scores of "first" "worst", or "multiple" were used and if a participant screened positive for any of the criteria (first, worst, and/or multiple), they were coded as TBI (1 = yes, 0 = no).

### 2.5. Social Network Variables: Social Capital

A social network interview was utilized during data collection [55]. Information collected were key independent variables. The survey involved a face-to-face social network interview conducted by a trained research staff member. The following prompt was first read: "Think about the past three months. Who are five people that you are closest to and have interacted and talked to (this could be face-to-face or over email, text, phone, social media, etc.) the most in the past three months." Participants were then asked to describe their relationship to each nominee: options consisted of the following: "friends from home or from before you were homeless; friends or other peers you know from the streets or peers you interact with at this agency; family (could be both biological and foster family); person you are romantically, intimately or sexually involved with; case workers, social worker, agency staff or volunteer; people from school; people from work; other." Participants were then asked, "who of the five people in your social network do you: talk to or see at least once per week; in the past 3 months: who have you spent time with, hung out with, chilled with, partied with, or had conversations with in person (i.e., face-to-face)?; interacted with via your phone, exchanged communication with via a tablet or computer; when you have been in crisis, feeling depressed or dealing with drama and major issues, who have you gone to for help or advice? (emotional support); who have you borrowed money or other material things from when you needed it? (instrumental support); who have you talked to about where to get social services (help with housing, food, clothes, casework, etc.; informational support)?" Social network data were exported from REDCap (Research Electronic Data Capture; ref. [92] and imported into SPSS for subsequent statistical analyses.

### Primary and Secondary Support Networks

Social networks can be divided into primary [88,89] and secondary networks [90], and this categorization is appropriate for individuals with a reported TBI [42]. Primary support networks consist of a romantic partner, family, and close friends, while secondary support networks consist of more formal and less personal relationships [42], including with friends or peers from the streets or with whom the participants interacted at this agency, caseworkers, and people from school or work.

Primary social networks were assessed by calculating the proportion of nominees who serve a primary support role (i.e., a romantic partner, family, or home-based peer) to total nominees. Secondary social networks were assessed by calculating the proportion of nominees who serve a secondary support role (i.e., street-based peers, caseworkers, or friends from school or work). We recoded categorical variables to address result skew-

ness [96,97]. The median could then be used to create a threshold for dichotomizing skewed variables [98]. The primary support variable was then dichotomized as either the presence of a nominee in the primary network (coded as 1) or the absence of a nominee in the primary network (coded as 0). Secondary support was dichotomized on the median to address the skewed distribution, similar to primary support. The secondary support variable was then dichotomized as either the presence of a nominee in the secondary network (coded as 1) or the absence of a nominee in the secondary network (coded as 0). Primary and secondary networks were then merged with types of support, including emotional, instrumental, and informational support. Therefore, social network variables were categorized as the following: primary network, secondary network; primary emotional support; primary instrumental support; primary informational support; secondary emotional support; secondary instrumental support; secondary informational support.

*2.6. Analytic Approach*

Data were exported and analyzed using SPSS (version 25; IBM Corp, 2017). To examine the relationship between social network variables and the dependent housing stability-substance use, bivariate logistic regressions were run to determine whether social capital variables were significantly associated ($p < 0.05$) with the dependent substance use variables. Only those social capital variables that were significant at the bivariate level were retained for the final multivariable logistic regression models in order to ensure statistical power and preserve degrees of freedom [99]. All demographic variables including race/ethnicity, gender, sexual orientation, educational attainment, and all TBI-related demographic variables including TBI (yes/no), first, worst, and multiple were retained for the multivariable logistic regression models. Listwise deletion was utilized for missing data because less than 10% of the data was missing.

## 3. Results

Sociodemographic, TBI-related variables, dependent variables, and all social capital characteristics are presented in Table 1.

**Table 1.** Descriptive characteristics of participants.

| Descriptive Characteristics of Participants (*n* = 115) | *n* (%) or M (SD) |
|---|---|
| Age (years) | 45.3 (13.3) |
| Gender | |
| Male | 75 (66.4) |
| Female | 38 (33.6) |
| Sexual Orientation | |
| Heterosexual | 98 (85.2) |
| Not Heterosexual | 17 (14.8) |
| Race and Ethnicity | |
| White | 73 (63.5) |
| Person of Color | 42 (36.5) |
| Education | |
| High school diploma or less | 70 (60.9) |
| More than high school diploma | 45 (39.1) |
| TBI Variables | |
| TBI total | 81 (70.4) |
| "Worst" injury | 53 (42.1) |
| "First" injury | 28 (22.2) |
| "Multiple" injury | 57 (45.2) |

**Table 1.** *Cont.*

| Descriptive Characteristics of Participants (*n* = 115) | *n* (%) or M (SD) |
|---|---|
| Substance Use Variables (dependent variables) | |
| Has your drinking or drug use led you to being kicked out of an apartment or program where you were staying in the past? (1 = yes). | 30 (26.3) |
| Will drinking or drug use make it difficult for you to stay housed or afford your housing? (1 = yes) | 15 (13.4) |
| Social Network Variables (*n* = 80) | |
| Nominated someone in primary support (family, partner, friend from before homelessness; 1 = yes) | 41 (51.2) |
| Secondary support | 39 (48.8) |
| Primary emotional support | 48 (60) |
| Primary instrumental support | 43 (53.8) |
| Primary informational support | 50 (62.5) |
| Secondary emotional support | 52 (65) |
| Secondary instrumental support | 49 (61.3) |
| Secondary informational support | 44 (55) |

### 3.1. Sociodemographic Characteristics

Out of 115 total participants, 66% identified as male and 34% identified as female.

The majority of participants, 85%, identified as heterosexual, and 64% identified as White. Additionally, 61% of participants had received a high school diploma or less and notably, 39% received more than a high school diploma.

### 3.2. Substance Use- Housing Stability

When examining housing instability related to substance use, 26% of participants reported that drinking or drug use had led them to be kicked out of an apartment or program where they were staying in the past. Additionally, 13% of participants reported that drinking or drug use would make it difficult for them to stay housed or afford housing.

### 3.3. TBI Demographics

Out of 115 total participants, 70% reported a significant history of TBI. The OSU TBI-ID screening showed that 42% of participants reported at least one head injury with a LOC for more than 30 min (worst). Additionally, 22% of participants reportedly experienced a TBI with LOC before the age of 15 (first). Moreover, 45% of participants reported experiencing either three or more head injuries, resulting in an altered state, or two or more TBIs with LOC within a 3-month period (multiple).

### 3.4. Social Capital Variables

#### 3.4.1. Primary and Secondary Support Networks

Social capital variables were categorized into two groups (primary and secondary support networks). Participants nominated a total of five people in their support network, and these each nominee was divided into a primary and secondary support role and then merged into types of support provided, including emotional support, instrumental support, and informational support. Overall, 51% of nominees were people who served primary support roles, meaning those who were categorized as family members, partners, or friends from before homelessness. Additionally, 49% of nominees were people who served secondary support roles, meaning street-based peers, caseworkers, or friends from school or work.

#### 3.4.2. Sources of Support: Emotional, Instrumental, and Informational Support

Primary and secondary support was then merged with various types of support provided including emotional support, instrumental support, and informational support. Of those nominees who were categorized into the primary support category, 60% pro-

vided emotional support, 54% provided instrumental support, and 63% provided informational support to the participant. More so, of those nominees who provided secondary support, meaning acquaintances rather than close family/friends, 65% provided emotional support to the participant, 61% provided instrumental support, and 55% provided informational support.

3.4.3. Social Network Composition: Differences between Participants with and without a Reported TBI

Differences in social network composition among individuals experiencing homelessness and a TBI compared to individuals experiencing homelessness only are displayed in Table 2. For example, 52% of participants with a reported TBI nominated someone in their primary network compared to 50% of participants without a reported TBI. As far as secondary networks, 48% of participants with a reported TBI nominated someone in their secondary network, compared to 50% of participants without a reported TBI. When examining network composition and types of support provided, 63% of participants with a reported TBI nominated someone who served a primary emotional support role while 54% of participants without a reported TBI nominated someone in their primary emotional support network. Regarding primary instrumental support, 56% of participants with a reported TBI nominated someone compared to 50% of participants without a reported TBI. As far as primary informational support was concerned, 63% of participants with a reported TBI nominated someone compared to 62% of participants without a reported TBI. When examining secondary emotional support, 63% of participants with a reported TBI nominated someone serving a secondary emotional support role compared to 69% of participants without a reported TBI. As far as secondary instrumental support was concerned, 59% of participants with a reported TBI nominated someone compared to 65% of participants without a reported TBI. Finally, regarding secondary informational support, 56% of participants with a reported TBI nominated someone compared to 54% of participants without a reported TBI.

**Table 2.** Differences in social network among individuals experiencing homelessness and a TBI compared to individuals experiencing homelessness only.

| Type of Support | TBI (1 = Yes) | No TBI (0 = No) |
|---|---|---|
| | *n* (%) | |
| Nominated someone in primary support (family, partner, friend from before homelessness; 1 = yes) | 28 (52) | 13 (50) |
| Secondary support | 26 (48) | 13 (50) |
| Primary emotional support | 34 (63) | 14 (54) |
| Primary instrumental support | 30 (56) | 13 (50) |
| Primary informational support | 34 (63) | 16 (62) |
| Secondary emotional support | 34 (63) | 18 (69) |
| Secondary instrumental support | 32 (59) | 17 (65) |
| Secondary informational support | 30 (56) | 14 (54) |

*3.5. Bivariate Findings*

As noted earlier in the analysis section, all demographic variables including race/ethnicity, gender, sexual orientation, and educational attainment, and all TBI-related demographic variables including TBI (yes/no), first, worst, and multiple, were retained for the multivariable logistic regression models as control variables. Social capital variables that were significant at the bivariate level were retained for the multivariable models. All secondary support variables (secondary emotional support, secondary instrumental support, and secondary informational support) were significant at the bivariate level and were therefore retained for the multivariable models.

### 3.6. Multivariable Findings

Multivariable models are presented in Table 3. There were two outcomes of interest (firstly, has your drinking or drug use led you to being kicked out of an apartment or program where you were staying in the past? Secondly, will drinking or drug use make it difficult for you to stay housed or afford your housing?) Significant findings for each model are reported.

**Table 3.** Multivariable Findings.

| | Has Your Drinking or Drug Use Led You to Being Kicked Out of an Apartment or Program Where You Were Staying in the Past? | | Will Drinking or Drug Use Make It Difficult for You to Stay Housed or Afford your Housing? | |
|---|---|---|---|---|
| Demographics | OR | 95% CI | OR | 95% CI |
| Race/Ethnicity | 2.6 | 0.59–11.34 | 0.18 * | 0.04–0.96 |
| Gender | 0.62 | 0.16–2.44 | 5.13 | 0.73–36.22 |
| Sexual Orientation | 2.84 | 0.28–29.01 | 0.53 | 0.07–4.24 |
| Education | 2.32 | 0.52–10.42 | 4.00 | 0.60–26.58 |
| TBI related variables | | | | |
| TBI (1 = yes) | 0.39 | 0.03–4.68 | 0.34 | 0.02–7.21 |
| First | 0.39 | 0.06–2.75 | 1.16 | 0.16–8.46 |
| Worst | 2.75 | 0.45–16.79 | 7.83 * | 0.96–63.29 |
| Multiple | 2.82 | 0.48–16.67 | 3.81 | 0.48–30.26 |
| Social network characteristics | | | | |
| Secondary emotional support | 45.85 ** | 4.61–455.94 | 2.65 | 0.32–22.20 |
| Secondary instrumental support | 0.08 * | 0.01–0.88 | 9.56 | 0.68–135.16 |
| Secondary informational support | 1.05 | 0.14–7.84 | 0.06 * | 0.00–0.98 |

Note. Reference category for: Has your drinking or drug use led you to being kicked out of an apartment or program where you were staying in the past? Secondly, will drinking or drug use make it difficult for you to stay housed or afford your housing? (0 = no); Race/ethnicity (BIPOC); gender minority; sexual orientation (sexual minority); less than high school education. Note. OR = Odds Ratio; 95% CI = 95% confidence interval. * $p < 0.05$. ** $p < 0.01$.

Participants who reported having been kicked out of an apartment or program due to drinking or drug use were more likely to have emotional support from a secondary network (OR = 45.85, $p < 0.01$, CI = 4.61, 455.94) and less likely to have instrumental support from a secondary network (OR = 0.08, $p < 0.05$, CI = 0.01, 0.88) compared to participants who had not been kicked out of an apartment or program due to drinking or drug use. Participants who said yes, drinking or drug use will make it difficult to stay or afford housing, were more likely to identify as White (OR = 0.18, $p < 0.05$, CI = 0.04, 0.96), were less likely to have reported informational support from a secondary network (OR = 0.06, $p < 0.05$, CI = 0.00, 0.98), and were more likely to have a reported "worst" TBI (OR = 7.83, $p < 0.05$, CI = 0.96, 63.29), compared to participants who reported that drinking or drug use will not make it difficult to stay or afford housing.

## 4. Discussion

Our study sought to provide answers to two research questions. The first research question investigated differences in social networks between people experiencing homelessness and a TBI and people experiencing only homelessness. We hypothesized that people experiencing both homelessness and TBI would have lower levels of social support overall than people experiencing only homelessness, due to the potential negative effect of TBI on social support [43]. The second research question examined the relationship between types (instrumental, emotional, informational support) and sources (primary and secondary network) of social support among individuals experiencing homelessness and their impact on housing stability related to substance use. We hypothesized that support from the primary network would be correlated with more housing stability while support from the secondary network would be associated with less housing stability. Several significant

findings came from this study that broaden our knowledge about social support among individuals experiencing homelessness and a TBI.

Findings showed high rates of reported TBI overall (70%), which is higher than some literature suggests [14]. This finding may further prove that individuals experiencing homelessness may be more likely to sustain a brain injury [22]. Alternatively, this finding may show that TBI may be a risk factor for homelessness [9,19,36,100]. Prevention strategies to prevent individuals with head injuries from experiencing homelessness is recommended. Additionally, interventions such as low barriers to housing services and rent supplements are recommended to improve one's living situation and provide safety.

Findings also revealed a lack of social support for participants with and without a reported TBI, highlighting the potential impact of homelessness and/or TBI on interpersonal relationships. Low levels of social support may result from homelessness; however, the unique vulnerabilities that lead to homelessness may also make these individuals vulnerable to low levels of social support, regardless of TBI status. Homelessness may be a symptom of other issues, such as mental health problems, physical health issues, and substance use [14,17,25,27] all of which can lead to a lack of social support. Additionally, overall rates of emotional support were higher than instrumental support regardless of source, suggesting that participants are more likely to have someone to meet emotional support needs than someone that provides tangible resources such as food, clothing, or shelter. This finding may suggest that individuals experiencing homelessness and a TBI connect with other individuals experiencing homelessness [53] who are also resource poor [55], and may be able to provide emotional support but not tangible help. It is recommended that efforts be made to connect individuals experiencing homelessness and TBI to home-based peers and family who may be able to provide instrumental support to help transition one out of homelessness.

Furthermore, support from the primary (51%) and secondary (49%) networks was comparable, implying that bridging social capital provided roughly half of overall support (primary network: family, partners, friends from before homelessness). This is a promising finding because previous research has linked bridging social capital to positive outcomes [65], such as increased housing resources, which may lead to improved health and housing stability [46,47]. Along with efforts to increase housing stability, intervention efforts may consider focusing on bridging social capital and building support networks.

While we expected to find lower levels of social support among people experiencing homelessness and a TBI compared to people experiencing only homelessness, our findings contradicted our hypothesis. Specifically, participants with a reported TBI nominated more people as sources of social support in a primary network (52% vs. 50%), higher rates of nominees from a primary network who provide emotional (63% vs. 54%) and instrumental support (56% vs. 50%), and nominees from a secondary network who provide informational support (56% vs. 54%) compared to participants without a reported TBI. One potential explanation is that individuals experiencing homelessness are often stereotyped and discriminated against [101–103], potentially leading to low levels of social support. Individuals with a reported TBI may be less stigmatized due to their physical and cognitive disabilities, and may therefore have higher levels of social support. More studies are needed to better understand why individuals with a TBI have higher levels of social support in this area.

Secondary support was associated with substance use related to housing stability in multivariable analyses. This finding may further show the role of secondary support specifically street-based peers on substance use behaviors. Previous research has found that longer periods of homelessness along with substance use behaviors were associated with increased engagement with street-based social networks [104]. It may be that substance use is a function of socialization with street-based peers who also use substances and this substance use may contribute to housing instability. Comprehensive service models that assess social networks among individuals experiencing homelessness and a TBI and

help these individuals establish and maintain connections to positive sources of support are recommended.

Additionally, participants who stated that their drinking or drug use would make it more difficult to maintain or afford housing were significantly more likely to report a TBI classified as "worst", defined as a TBI with LOC lasting more than 30 min [95]. This finding may imply that in addition to the direct consequences of TBI, individuals may experience unstable housing and substance use as individuals struggle to manage and cope with the functional impact of their TBI [26]. Case management and rent supplements are recommended first to improve one's living situation and safety [105] followed by substance use treatment and treatment for TBI.

Individuals who reported that substance use would make it difficult to stay or afford housing were also significantly less likely to have reported secondary informational support than participants whose housing was unaffected by substance use. Substance use is complex on its own, and when combined with TBI, it presents a number of challenges to TBI recovery and social support. TBI may contribute to lower levels of informational support due to the consequences of TBI, such as poor planning and organization [106]; individuals with a TBI may find it more difficult to remember appointments with case workers or to complete paperwork for medical care or housing services. To follow a harm reduction approach, low barrier and flexible treatment models should be recommended for housing services and case management.

Participants who indicated that they had been kicked out of housing in the past due to drinking or drug use, on the other hand, were more likely to have secondary emotional support than participants who had not been kicked out of housing in the past due to substance use. Emotional support from peers on the streets can foster important relationships as well as a sense of community and familiarity. According to research, individuals who do not have stable housing and then move to stable housing frequently have to leave a familiar street culture [107]; perhaps a familiar environment with emotionally supportive peers is more important to some than stable housing.

Additionally, drinking or drug use leading to being kicked out of housing in the past was associated with less likelihood of secondary instrumental support, meaning someone from whom the participants had borrowed money or other material things when they needed it. Perhaps substance use is more of a function of socialization with peers who also use substances; these substance-using peers provide companionship and emotional support instead of instrumental support [55]. Interventions that help establish and maintain instrumental support through caseworkers or through the primary network are recommended.

*Limitations*

Study limitations should be noted. Cross-sectional data were used for this study, limiting causal conclusions. Utilizing longitudinal data in future studies may clarify the causal relationships between homelessness, TBI, social networks, substance use, and housing stability. Furthermore, data were self-reported and could be biased due to the sensitive topics asked of the participants. Another limitation is that participants were recruited from service agencies serving adults experiencing homelessness, and so the sample is likely not representative of all adults experiencing homelessness, especially those who are disconnected from services. Measurement limitations were that substance use-related variables came from the VI-SPDAT, a measure designed to screen for housing assistance. Future studies should consider using validated substance use measures to assess the impact of TBI and homelessness on substance use related to housing stability. In addition, this study was limited geographically to two cities in Colorado; future research should consider using samples from diverse areas for more generalizable findings. Additionally, severity of substance use was not measured and should be considered for future research. Lastly, this study only included English-speaking adults. Researchers should consider administering studies in multiple languages in the future.

## 5. Conclusions

Our study findings demonstrate that social support among adults experiencing homelessness is low regardless of TBI status. While we expected to find differences in social network composition and types of support provided between individuals experiencing homelessness with a TBI and without a TBI, there is more commonality in the rates of reported social support. While our findings are preliminary, they offer important implications for interventions among individuals experiencing homelessness and a TBI. For example, research has shown that the combination of rent supplements and intensive case management led to greater housing stability and increased social networks for veterans experiencing homelessness with psychiatric or substance use disorders [108]. Perhaps low barriers to permanent supportive housing along with intensive support will yield positive results for adults experiencing homelessness with a TBI.

**Author Contributions:** Conceptualization, S.C., G.S., S.C.R., K.C., B.B., K.G. and D.B.; Formal analysis, S.C. and A.B.-A.; Investigation, S.C. and A.B.-A.; Methodology, K.G. and D.B.; Project administration, S.C., K.G. and D.B.; Supervision, K.G. and D.B.; Writing—original draft, S.C., K.C., B.B., S.C.R. and G.S.; Writing—review & editing, S.C., K.C., B.B., S.C.R., A.B.-A., K.G. and D.B. All authors have read and agreed to the published version of the manuscript.

**Funding:** This research was funded by the University of Denver Professional Research Opportunity for Faculty. And by Mindsource Brain Injury Network in Colorado.

**Institutional Review Board Statement:** The study was conducted in accordance with the University of Denver, and approved by the Institutional Review Board of University of Denver (1521142-9 and 19 October 2021).

**Informed Consent Statement:** Informed consent was obtained from all subjects involved in the study.

**Data Availability Statement:** Not applicable.

**Conflicts of Interest:** The authors report no conflict of interest.

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
