# Peer review of "Substance Use and Housing Stability among Individuals Experiencing Homelessness and a Traumatic Brain Injury: The Role of Social Support"

_psychoactives, doi:10.3390/psychoactives2020010_

Round 1
Reviewer 1 Report
This study explores interconnections between TBI, homelessness and substance use. The literature review covers several heavily studied topical areas that are linked (homelessness, substance use, mental illness) as they interact with and possibly exacerbate TBI. Some sections are not necessary (or can be shortened) as the focus is on TBI, e.g., section 1.2.1 Homelessness and Substance Use, 1.3 on Social Support and Homelessness, 1.6.1 on Social Capital theory.
I assume all of the empirical studies reported on are correlational and thus cannot show causation.
On pp. 5-6, the authors conflate social networks, social support and social capital when these concepts refer to distinct (though overlapping) phenomena (social networks are sociometric displays of individuals who are linked with the ‘ego’, social support indicates emotional or instrumental assistance, while social capital refers to resources and goods that may be available and ‘activated’ through a relationship (bonding, bridging or linking). The mis-labeling occurs again in section 3.4. This raises the question of whether social capital ‘theory’ is appropriate for this study and how its influence can be detected. In any event, the necessity of measuring and analyzing all three indicators interchangeably (social support, social capital, social network) is not clear and not rationalized.
Hypothesis 1 fits the goals of the study. The second question/hypothesis focuses on housing security and social support (TBI is not included). How is the 2nd questions related to the study’s basic goal? The primary dependent variables (answers to 2 questions about substance use as a cause of housing instability) do not appear relevant given the primary target of interest (TBI).
How and why was ‘purposive’ sampling used? This sampling technique is associated with qualitative studies. Was TBI part of the inclusion criteria (otherwise a rate of 70% appears very high). It would help to resolve the chicken-or-egg question of what came first (TBI or homelessness).
Why is 2.4.1 section needed when demographic characteristics are enumerated in a table and need only a summary narrative? Similar concerns about redundancy appear in Table 2 and its description---when visually displayed, one does not need to describe the table’s contents in detail. Were the comparisons in Table 2 subjected to statistical analyses of differences between TBI and non-TBI group?
The study’s findings are not surprising and do not lead to clear recommendations given the many variables included in the model and the lack of rationale beyond an under-specified
theory (social capital) for the predictions made.
Reviewer 2 Report
- The study has purposive sampling which was required to identify subset population in question. Appropriate data gathering scales used. No methodological inaccuracies noted as such keeping in mind nature of sample and variables in question. Agree with author hypothesis.
- I feel this article gives appropriate review, has successfully tried to fill the gap in knowledge. Appropriate references have been cited.
- I would like to ask the authors if they considered severity of substance use (more severity leads to more contact with law enforcement, potentially more hospital visits, so in a way actually leading to more resources/support in the form of outpatient clinical referrals/case management etc. for those folks. Consider adding that as limitation if appropriate.
- Similarly other co-morbid medical issues that lead to hospital visits and therefore more primary/secondary support because of that could be confounding as more support. This potentially could be limitation in making conclusions as this was not screened. This fact was notable in this study as TBI patients with homelessness identified more supports than just homeless.
